# Site-specific identification and quantitation of endogenous SUMO modifications under native conditions

Ryan J. Lumpkin[1], Hongbo Gu[2], Yiying Zhu[2], Marilyn Leonard[3], Alla S. Ahmad [1], Karl R. Clauser[4], Jesse G. Meyer [1], Eric J. Bennett [3] & Elizabeth A. Komives[1]

Small ubiquitin-like modifier (SUMO) modification regulates numerous cellular processes. Unlike ubiquitin, detection of endogenous SUMOylated proteins is limited by the lack of naturally occurring protease sites in the C-terminal tail of SUMO proteins. Proteome-wide detection of SUMOylation sites on target proteins typically requires ectopic expression of mutant SUMOs with introduced tryptic sites. Here, we report a method for proteome-wide, site-level detection of endogenous SUMOylation that uses α-lytic protease, WaLP. WaLP digestion of SUMOylated proteins generates peptides containing SUMO-remnant diglycyl-lysine (KGG) at the site of SUMO modification. Using previously developed immuno-affinity isolation of KGG-containing peptides followed by mass spectrometry, we identified 1209 unique endogenous SUMO modification sites. We also demonstrate the impact of proteasome inhibition on ubiquitin and SUMO-modified proteomes using parallel quantitation of ubiquitylated and SUMOylated peptides. This methodological advancement enables determination of endogenous SUMOylated proteins under completely native conditions.

[1] Department of Chemistry and Biochemistry, University of California, San Diego, La Jolla, CA 92093-0378, USA. [2] Proteomic Service Group Cell Signaling Technology, 3 Trask Lane, Danvers, MA 01923, USA. [3] Cell and Developmental Biology, University of California, San Diego, La Jolla, CA 92093-0380, USA. [4] Broad Institute of MIT and Harvard, 415 Main St, Cambridge, MA 02142, USA. Eric J. Bennett and Elizabeth A. Komives contributed equally to this work. Correspondence and requests for materials should be addressed to E.J.B. (email: e1bennett@ucsd.edu) or to E.A.K. (email: ekomives@ucsd.edu)

The family of small ubiquitin-like modifier (SUMO) proteins in humans includes four distinct genes with three types of members: SUMO1, SUMO2/3 (which differ by only three residues), and SUMO4. SUMO proteins regulate the function of various proteins by reversible covalent isopeptide bond attachment between the C terminus of SUMO and a free ε-amine group typically on lysine residues within target proteins[1], similar to ubiquitin (Ub). Ub conjugation mainly targets proteins for degradation by the proteasome, but has also been implicated in DNA repair, receptor signaling, and cell communication[2]. The function of SUMO conjugation on target proteins is similarly diverse with SUMOylation catalyzing alteration of protein activity for targets involved in gene expression, DNA repair, nuclear import, heat shock, cell motility, and lipid metabolism[1,3]. SUMO targets are generally low-abundance proteins, and the amount of the modification at steady state is also low[4]. Given its importance in numerous cellular functions, several groups have developed proteomic methods for analysis of SUMOylated proteins.

Three general approaches have been previously utilized to isolate and identify the SUMOylated proteome[5]. Ectopic expression of epitope-tagged SUMO followed by standard isolation techniques and mass spectrometry has been widely used in a variety of organisms[6–10]. These approaches have yielded various maps of SUMO-interacting proteins but few sites of SUMOylation are identified by this approach and the extent to which exogenous expression of modified SUMOs alters substrate targeting is unknown. Immuno-affinity approaches utilizing antibodies that recognize endogenous SUMO2/3 or SUMO1 have been used to identify SUMO-interacting proteins under endogenous conditions[4]. However, as with the epitope-tagging approach, few actual sites of SUMO modification were identified using this method. Therefore, methods that allow proteome-level identification of endogenous SUMOylation sites are needed.

A robust proteomic method has been developed to measure thousands of endogenous ubiquitylation sites[11]. The method takes advantage of the C-terminal sequence of Ub (RGG) (Fig. 1a). When cleaved with trypsin, ubiquitylated substrate proteins will generate peptides containing a Ub-remnant diglycyl-lysine (KGG) that can be enriched using specific antibodies and identified by tandem mass spectrometry (Fig. 1b)[12–14]. Instead of the trypsin-friendly arginine residue preceding the C-terminal diGlycine sequence observed in the processed Ub sequence, mature human SUMO paralogs have a threonine preceding the C-terminal diGlycine sequence and no other tryptic cleavage sites near the C terminus (Fig. 1a). To overcome this problem, various schemes that introduce mutations within the C terminus of SUMO to render it more amenable to trypsin-based cleavage and identification by mass spectrometry have been developed for global profiling of SUMO attachment sites[15,16]. Several groups have reported global profiling approaches in which mutant SUMOs with various affinity tags and protease recognition sites were introduced into cells. For example, Hendriks et al. introduced a lysine-deficient SUMO-3 with a C-terminal trypsin cleavage site, $His_{10}$–SUMO-3 K0 Q87R[17,18]. The SUMOylated proteins were enriched by immobilized metal affinity chromatography (IMAC) and digested with Lys-C. Peptides modified with SUMO were then purified again with IMAC and finally digested with trypsin to generate a five amino acid C-terminal SUMO-remnant modification. This group has compiled all available data resulting in 7327 SUMOylation sites in 3617 proteins[17]. Other groups have used similar engineering approaches to express mutant SUMOs and identify up to 1000 unique SUMOylation sites upon induction of cell stress[15,16,18–20]. While these methods have proved effective in mapping SUMOylation sites, they all require exogenous expression of mutant version of SUMO, which

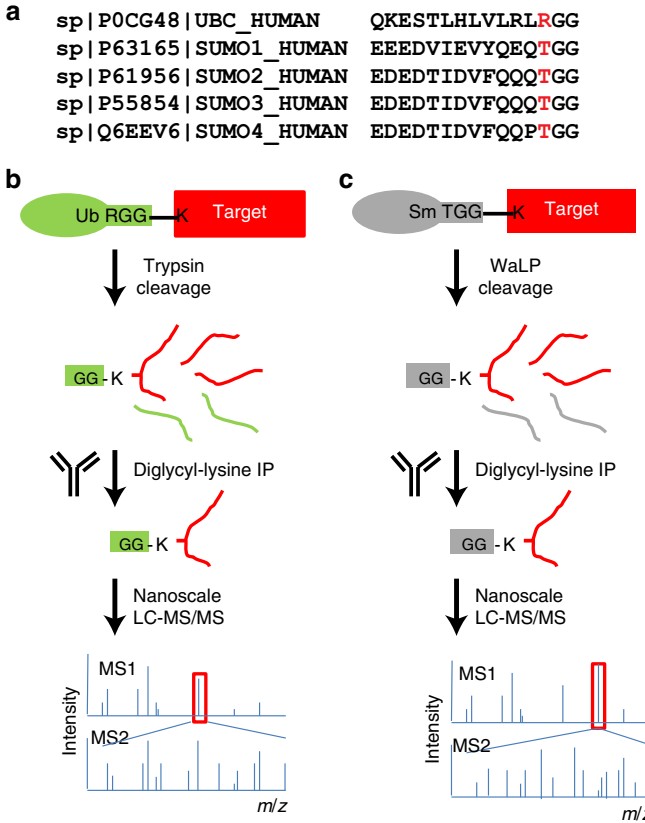

**a**

| sp\|P0CG48\|UBC_HUMAN | QKESTLHLVLRL**R**GG |
| sp\|P63165\|SUMO1_HUMAN | EEEDVIEVYQEQ**T**GG |
| sp\|P61956\|SUMO2_HUMAN | EDEDTIDVFQQQ**T**GG |
| sp\|P55854\|SUMO3_HUMAN | EDEDTIDVFQQQ**T**GG |
| sp\|Q6EEV6\|SUMO4_HUMAN | EDEDTIDVFQQP**T**GG |

**Fig. 1** A strategy for mapping endogenous SUMO-modification sites. **a** C-terminal sequence alignment of processed human SUMO1-4 and ubiquitin. The proximal amino acid to the diGlycine C-terminal residues is indicated in red. **b** Schematic depicting the Ub site mapping strategy. Proteins modified by ubiquitin are digested with trypsin leaving a diGlycine attached to the ε-amine of the lysine where ubiquitin was attached. An antibody specific for KGG-peptides is used to enrich peptides from ubiquitylated sites that are then identified by mass spectrometry. **c** The same as **b** but WaLP digestion is used to generate the KGG-peptides from SUMO attachment sites

preclude analysis of SUMO-modification sites in native settings or from human tissues. Currently, no method exists for identifying endogenous SUMO sites on a global proteome scale without introduction of mutant SUMO.

We recently described the application of wild-type α-lytic protease (WaLP) to proteome digestion for shotgun proteomics[21]. Although relatively relaxed specificity was observed, WaLP prefers to cleave after threonine residues and rarely cleaved after arginine[21]. In addition, WaLP generates peptides of the same average length as trypsin despite its more relaxed substrate specificity[21]. We show here that WaLP cleaves at the C-terminal TGG sequence (all SUMO paralogs) leaving a SUMO-remnant KGG at the position of SUMO attachment in target proteins. The resulting KGG-containing peptides can then be identified using methods already developed for profiling the Ub-modified proteome as described above (Fig. 1c). The method allows identification of SUMO attachment sites under completely native conditions using the Ub-profiling workflow by simply substituting WaLP for trypsin. The same sample can be subjected to analysis of both the Ub- and SUMO-modified proteomes simply by digesting the sample with either trypsin or WaLP, respectively. We demonstrate the effectiveness of this parallel identification of ubiquitylation and SUMOylation sites in cells treated with

proteasome inhibitors. We provide the description a unique method of identifying proteins containing individual lysine residues that are modified by both SUMO and Ub from the same sample. This method can be simply applied to any sample, including human tissues samples, to identify endogenous SUMOylation sites.

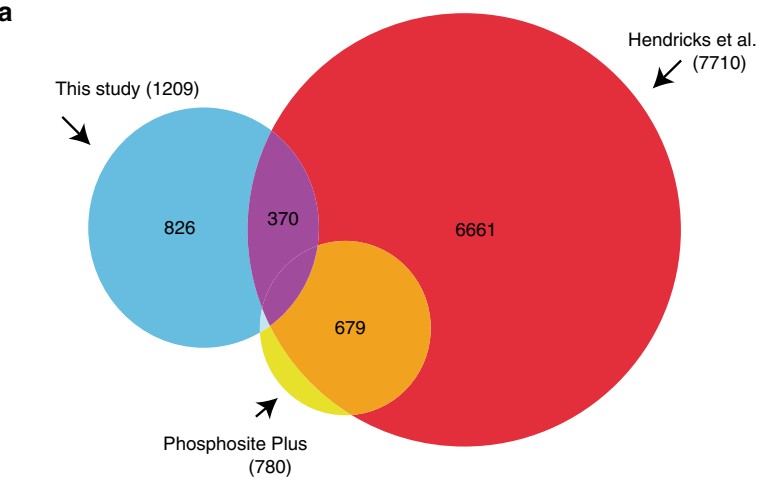

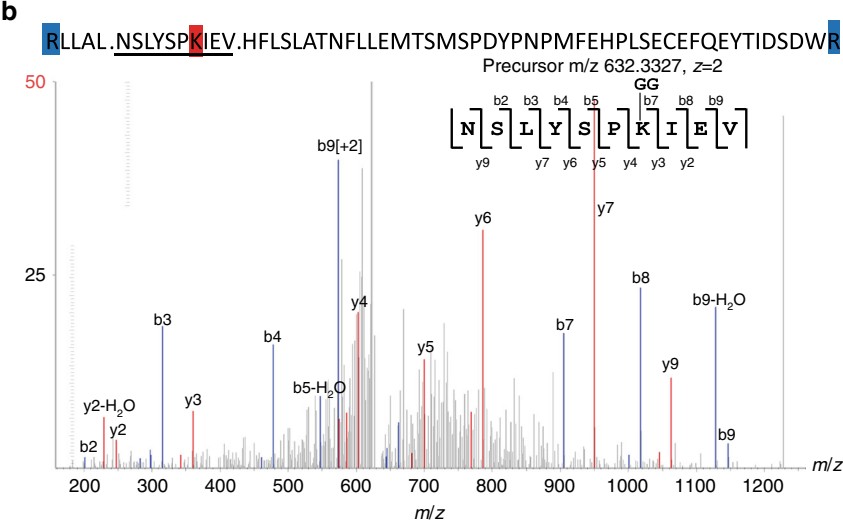

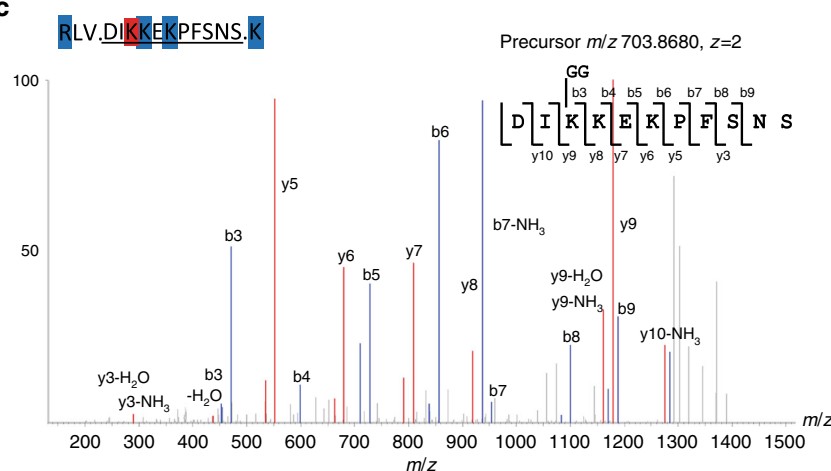

**Fig. 2** Summary of SUMOylation site identifications from this study and comparison with previous studies. **a** Venn diagram showing overlap between the 1209 SUMO sites identified in this study from both Hela and HCT116 cell lines (blue) and all SUMO modifications from either Hendriks et al. (red) or Phosphosite Plus (yellow). **b** Identified spectrum of a novel SUMO modification site identified in DNA-PK that would have resulted in a 57-amino-acid tryptic peptide. The sequence above the spectrum shows the tryptic cleavage sites in blue, the modification site is in red, and the matched sequence from the WaLP digest is underlined. **c** Identified spectrum of a previously uncharacterized SUMO modification site in Sp100 that would result in only a six-residue tryptic peptide

## Results

**SUMOylation site profiling by WaLP digestion.** Previous studies demonstrated the utility of WaLP digestion in shotgun proteomics platforms[21]. Our observation that threonine was among the preferred amino acids in the P1 position for WaLP digestion[21] led us to hypothesize that digestion of SUMOylated proteins with WaLP, which should cleave after the threonine in the SUMO C-terminal sequence, TGG, would generate a SUMO-remnant diglycyl-lysine (KGG) at SUMOylation sites (Fig. 1a). The same workflow used for Ub-remnant profiling could then be used to globally profile SUMO attachment sites (Fig. 1b, c). Although WaLP can simply be substituted for trypsin during sample preparation, identification of non-tryptic peptides produced from WaLP digestion is challenging because search engines score on the basis of b and y ion series that are expected from tryptic peptides with a C-terminal positive charge. We previously demonstrated that identification of peptides arising from WaLP digestion benefits from electron transfer dissociation (ETD)[22], whereas higher-energy collisional dissociation generates internal ions that can complicate peptide spectral matching[21]. WaLP cleaves after at least four different amino acids requiring the use of "no enzyme" specificity in database searches, which is challenging for many publicly available search algorithms. To facilitate mapping of SUMOylation sites using WaLP digestion, we used the MS-GF+ search engine whose scoring function can be trained using identifications from an initial search[23,24].

**Identification of novel SUMOylation sites using WaLP.** As an initial test of the method, we digested lysates generated from human cell lines with WaLP and enriched for KGG-containing peptides using established protocols for Ub-modified peptide enrichment[14]. Subsequent analysis by mass spectrometry and database searching using MS-GF+ resulted in the identification of 2051 unique KGG-containing peptides, which were confidently localized (PTMprophet score >0.9) to 1209 unique sites (Supplementary Data 1). Comparison of the SUMOylation sites identified using our WaLP digestion approach with the SUMOylation sites amalgamated by Hendriks et al.[17] revealed an overlap of only 30% (Fig. 2a). When compared to previously reported SUMOylation sites from Hendriks et al. (7710 sites)[17], Phosphosite Plus (780 sites)[25], and Uniprot (1863 sites)[26], 826 were novel. Peptides containing SUMO remnants from WaLP digestion may correspond to sequences that are not covered by tryptic digestion due to the abundance or lack of nearby tryptic cleavage sites[21]. We found several examples of such cases. For example, we identified a novel SUMOylation site in DNA-PK (PRKDC, P78527), which would have been in a tryptic peptide of length 57 (Fig. 2b), and another in Sp100 (P23497), which would have been a tryptic peptide of only six amino acids (Fig. 2c). Multiple SUMOylation sites were found in over 60% of the proteins identified, with 5% having more than six SUMOylation sites. The SUMOylation sites identified in this study correspond to the expected motifs as described by others. We found that 31% of the SUMO attachments occurred at the "forward" sequence motif (ΨKX[E|D]) (Supplementary Fig. 1a), 9% corresponded to the "inverted" motif ([E|D]XK) (Supplementary Fig. 1b), and 60% did not correspond to either consensus but these sites were somewhat enriched in acidic residues (Supplementary Fig. 1c). Gene ontology term enrichment was performed to functionally annotate the SUMO-modified proteins. As reported previously, SUMOylation sites were primarily found in nuclear proteins and SUMOylated proteins were enriched for proteins involved in chromatin biology, RNA metabolism, and transcription (Supplementary Fig. 2).

It is possible that the same lysine within an individual protein can be either modified by Ub or SUMO and that SUMOylation may antagonize ubiquitylation[27–29]. In fact, 30% of the sites identified in this study were either previously reported to be ubiquitylated in the Phosphosite Plus database or were found in our experiments to be ubiquitylated. This overlap is similar to that previously reported by Hendriks et al.[18].

**Validation of SUMOylation sites using in vitro deSUMOylation.** Our previous studies on WaLP digestion specificity indicated that WaLP rarely cut after arginine residues (Fig. 1b) leads to the possibility of using WaLP to generate diGlycine-remnant peptides from SUMO-modified proteins and not Ub-modified proteins. However, when using WaLP to isolate and identify SUMOylated peptides in this study, we occasionally identified peptides whose MS/MS spectra matched best to peptides that resulted from cleavage after arginine either on the N or C terminus. These peptides were rare, with percentages varying from 3–8%, which we attribute at least partly to our inability to completely remove the trypsin used to detach cells during scale-up. Trypsin was avoided in cell harvesting, scraping was used instead. Even still, the fact that the P1 Arg was usually present in a sequence that had multiple positively charged residues in a row led us to question whether WaLP digestion could occasionally result in the generation of a KGG-peptide from a ubiquitylated protein. To evaluate this possibility and to validate our approach, native cell lysates were either untreated or treated with recombinant deSUMOylating enzymes, SENP1 and SENP2, prior to digestion with either trypsin or WaLP. Western blot analysis revealed that in vitro treatment with SENP1/2 resulted in a robust reduction of SUMO1- and SUMO2/3-modified proteins while leaving ubiquitylated proteins unaltered (Fig. 3a). Digestion of SENP1/2-treated lysates with trypsin or WaLP and subsequent analysis by mass spectrometry revealed a dramatic reduction in the abundance of KGG-modified peptides from WaLP-digested samples compared to untreated samples (Fig. 3b). Importantly, the abundance of KGG-peptides resulting from trypsin cleavage, which would arise from ubiquitylated substrates, was unaltered by SENP1/2 treatment (Fig. 3b). The observation that 88% of sites identified after WaLP digestion were decreased at least twofold upon SENP1/2 treatment validated our approach. Conversely, treatment of lysates with a promiscuous deubiquitylating enzyme, Usp2cc, would result in a specific decrease in the amount of a KGG-peptide from a ubiquitylated site without altering SUMOylated peptides. Consistent with our prediction, less than 2% of KGG-peptides generated from WaLP-digested cells decreased upon Usp2cc treatment (a similar percentage to the FDR), whereas 97% of the KGG-containing peptides generated from trypsin digestion decreased after Usp2cc treatment (Fig. 3c). Taken together, we conclude that WaLP digestion and subsequent immuno-affinity enrichment of KGG-modified peptides specifically identifies endogenous SUMOylated peptides.

**Parallel Ub and SUMO site identification.** A unique feature of our method is the ability to identify ubiquitylation and SUMOylation sites in parallel from the same cell or tissue lysate. The lysate can be simply split in half, with one half digested with WaLP and the other half digested with trypsin (Fig. 1c, d). The subsequent immuno-affinity enrichment steps are identical, although mass spectrometry and data processing were optimized for tryptic peptides for Ub and non-tryptic peptides for SUMO. As is already well established, SUMOylation is a lower abundance modification than ubiquitylation, and indeed, we identified 6472 Ub sites in the same samples in which 1209 SUMO sites were identified. Qualitative comparisons of the data revealed all

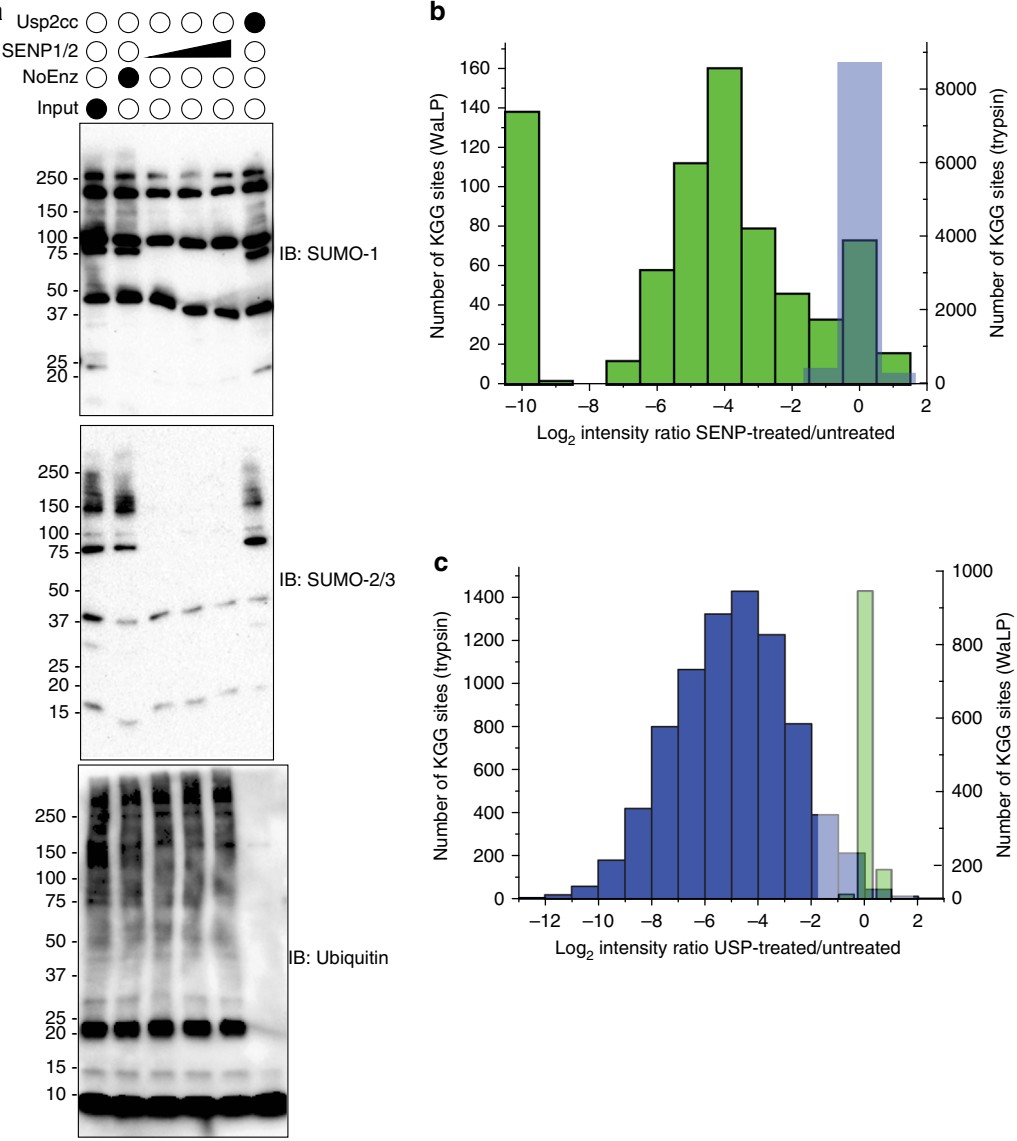

**Fig. 3** Identification and quantitation of SUMO and Ub modification sites upon in vitro deSUMOylation in Hela cell extracts. **a** Extracts were either untreated or treated with varying amounts of SENP1 and SENP2 enzymes or Usp2cc and then analyzed by SDS-PAGE and blotted for SUMO1 or SUMO2/3 or ubiquitin. **b** Bar graph showing results from quantitation of KGG-peptides after treatment with SENP1/2, illustrating the reduction of KGG sites observed after WaLP digestion. Green bars indicate SUMO modification sites identified and quantified upon WaLP digestion (left, y axis) and blue bars indicate Ub modification sites identified and quantified upon trypsin digestion (right, y axis). **c** Bar graph showing results from quantitation of KGG-peptides after treatment with Usp2cc, illustrating the retention of KGG sites observed after WaLP digestion. Colors are the same as in **b**

possible types of dual modification scenarios. We found 2713 proteins that were ubiquitylated, 768 proteins that were SUMOylated, and 407 proteins that had both SUMO and Ub. Some had a large number of ubiquitylated lysines and few SUMOylated lysines while others had larger numbers of SUMO-modified lysines and few ubiquitylated sites. Those for which we observed both modifications mostly carried those modifications on different residues. We observed 243 lysines that were found to be Ub modified in the trypsin-digested samples and SUMO modified in the WaLP-digested samples. These results indicate that our method can identify proteins that are modified by both SUMO and Ub on the same lysine residues in the same sample.

**Parallel quantification of Ub and SUMO sites**. Previous studies indicated that SUMO modification could be stimulated by cell

stresses such as heat shock and proteasome inhibition[30,31]. Using SILAC-based quantitative proteomics, and our method allows for the parallel capture, identification, and quantification of SUMO- and Ub-modified proteins in response to various cell perturbations. As it is well established that proteasome inhibition results in global alteration of the Ub-modified proteome, we evaluated the dynamic response of both the Ub- and SUMO-modified proteome to proteasome inhibition. Metabolically labeled cells were treated with MG132 and mixed with unlabeled cells prior to cell lysis. Again, the cell lysates were split and half was digested with trypsin for Ub analysis and the other half was digested with WaLP for SUMO analysis. We identified and quantified 330 unique SUMOylated sites and 2621 unique ubiquitylation sites from biological replicate samples. The results for both experiments were normally distributed (Fig. 4) and the results from the Ub analysis were comparable to previously published data in

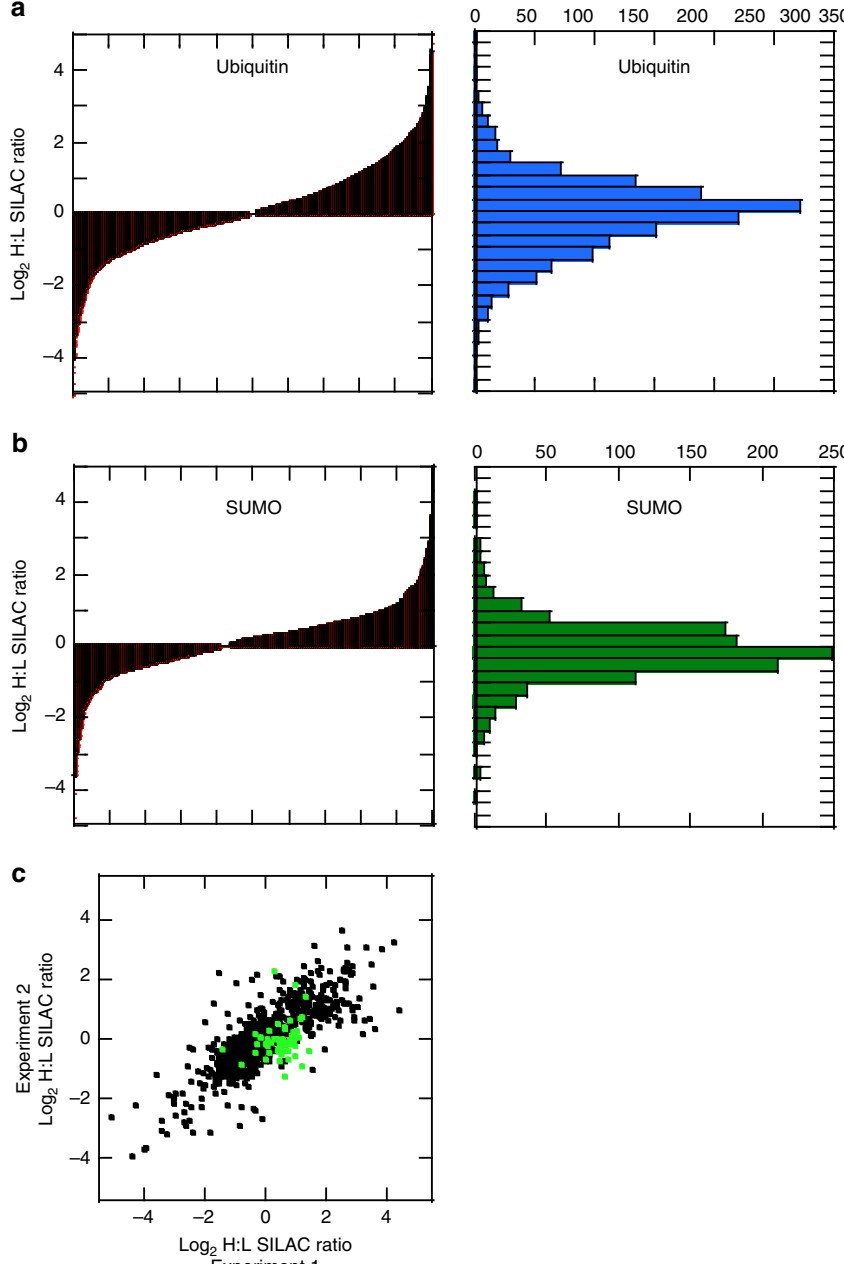

**Fig. 4** Parallel identification and quantitation of the SUMO and Ub-modified proteome upon MG132 treatment of HCT116 cells. Plot of normalized log(2) SILAC ratios (heavy/light) for all quantified unique Ub modification sites **a** or SUMO modification sites **b** upon MG132 treatment. The left graphs depict the ordered distribution of the SILAC ratios and right graphs depict the histogram of SILAC ratios. **c** Comparison of the SILAC ratios from biological replicates for ubiquitylation sites (black symbols) or SUMOylation sites (green symbols)

which proteasome inhibitors were utilized[14,32]. For the SUMO analysis, 30 sites increased in abundance by at least twofold and 72 sites decreased in abundance by at least twofold upon MG132 treatment[18]. Changes in modification abundance are not likely due to changes in protein levels as protein levels were not found to change significantly during a 4-h MG132 treatment[14]. We observed 103 proteins for which both SUMOylation and ubiquitylation could be quantified. These proteins had a total of 591 modified lysines. Of the total 591 modified lysines, 53 were found to be both SUMOylated and ubiquitylated.

The proteins with the most abundant sites that were both SUMOylated and ubiquitylated were SUMO and Ub themselves,

so we analyzed these to see if any sites were reciprocally regulated. It is important to note that we observed multiple modified peptides with more than one SUMO modification on SUMO proteins only in the WaLP digest. This indicates the presence of SUMO molecules that are simultaneously SUMOylated at different lysine residues and may have a branched-chain architecture. Interestingly, SUMO-modified ubiquitin was only observed on single lysine residues, suggesting that multiply SUMO-modified ubiquitin is a rare or nonexistent event. Conversely, peptides with more than one Ub modification on ubiquitin were observed only in the trypsin digest. These results are consistent with the presence of poly-SUMO (detected by

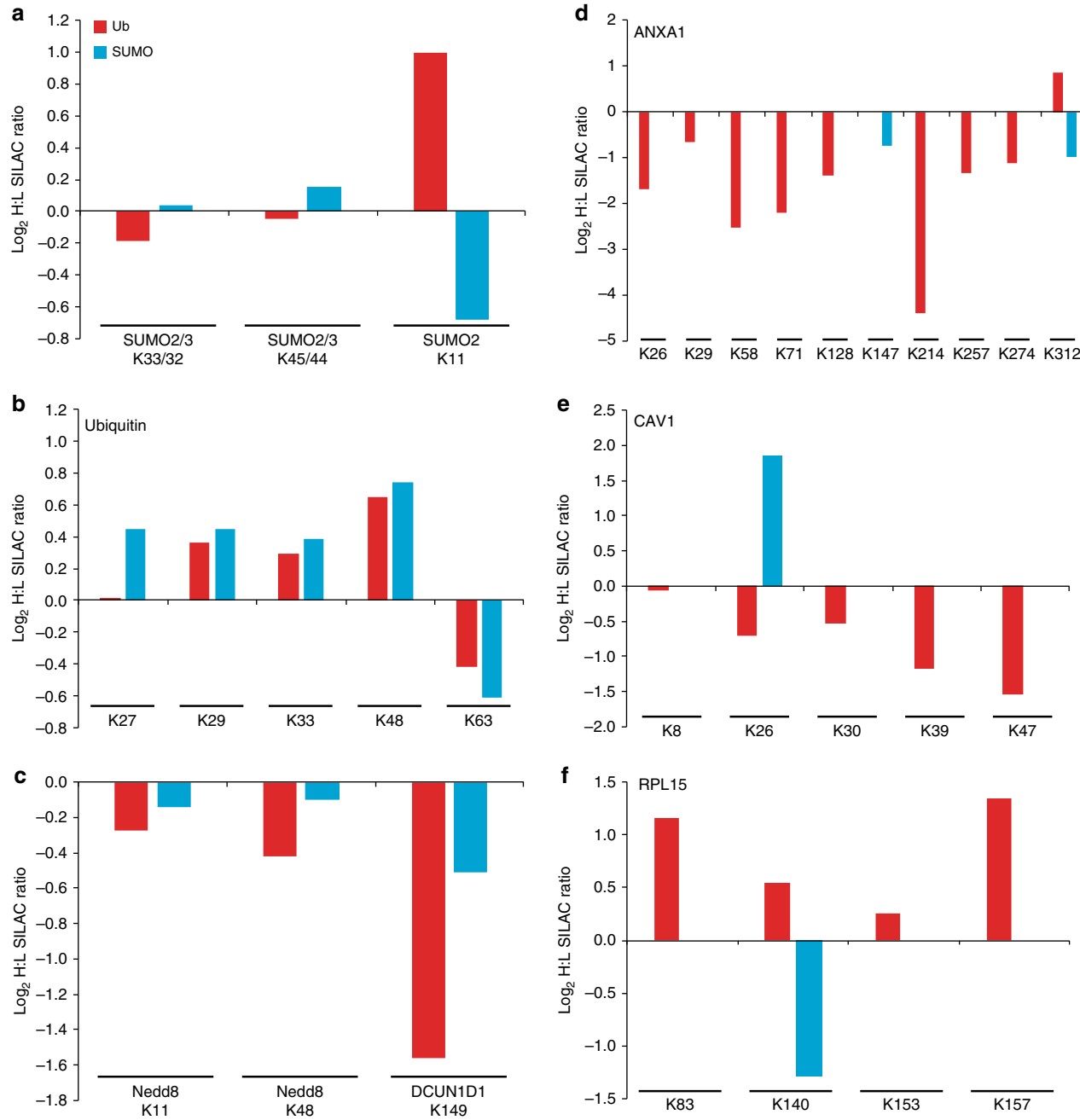

**Fig. 5** Log$_2$ SILAC ratios (heavy-MG132-treated:light-untreated HCT116 cells) for quantified ubiquitin (red) and SUMO (blue)-modified peptides observed on SUMO-2/3 (**a**), ubiquitin (**b**) Nedd8 and DCN1 (**c**), annexin-1 (**d**), caveolin-1 (**e**), or RPL15 (**f**). The position of the modified lysine is indicated

WaLP) and poly-Ub (detected by trypsin) but only mono-modifications of the opposite protein. We quantified ubiquitylation and SUMOylation of three lysine residues in SUMO2/3 upon MG132 treatment (Fig. 5a). Neither ubiquitylation nor SUMOylation of lysines 33 or 45 in SUMO2/3 were dramatically altered upon proteasome inhibition. However, ubiquitylation of lysine 11 on SUMO2/3 increased, whereas SUMOylation on this site decreased upon MG132 treatment (Fig. 5a). Cross-modification of SUMO2/3 by ubiquitin has been previously observed, and consistent with these results, ubiquitylation of SUMO2/3 on lysine 11 increases upon proteasome inhibition[31].

For ubiquitin, we were able to quantify Ub and SUMO modification of lysines 27, 29, 33, 48, and 63 (Fig. 5b). Both

SUMOylation and ubiquitylation of ubiquitin responded similarly to MG132 treatment (Fig. 5b). Both SUMO and Ub modification of lysines 48, 29, and 33 increased in abundance upon MG132 treatment. Conversely, both SUMO and Ub modifications on lysine 63 of Ub decreased in abundance in the MG132-treated cells. Similarly, we observed cross-modification of another ubiquitin-like (Ubl) protein, Nedd8. Lysines 11 and 48 on Nedd8 were observed to be modified by both Ub and SUMO and the extent of modification was largely unperturbed by proteasome inhibition (Fig. 5c). Trypsin digestion followed by KGG immuno-affinity enrichment cannot distinguish between ubiquitylation or neddylation due to the presence of an arginine preceding the C-terminal diGlycine in both Ub and Nedd8. As such, it is possible

that Nedd8 may be poly-neddylated at these sites rather than ubiquitylated. Consistent with this observation, DCN1 was observed to be diGlycine-modified at a lysine residue previously characterized to be a neddylation site and this same lysine residue was SUMOylated (Fig. 5c). Interestingly, the abundance of both SUMOylated and neddylated DCN1 decreased upon proteasome inhibition. These observations suggest that cross-modification of Ubl proteins is prevalent and may possibly be antagonistic modifications.

Separate from the highly abundant Ubl proteins, we observed a few examples of proteins that were modified by both SUMO and Ub on the same lysine residue with differential responses to proteasome inhibition. For example, we quantified two SUMOy-lation and nine ubiquitylation sites on annexin A1 (Fig. 5d). We observed SUMOylation at Lys 147, which has not been previously reported, and at Lys 312, a known SUMOylation and ubiquityla-tion site[18]. This C-terminal lysine appears to experience reciprocal regulation by MG132 as the abundance of the SUMO-modified form was significantly decreased, whereas the ubiquitylation was significantly increased (Fig. 5d). Further, lysine 26 in caveolin-1 and lysine 140 in the large ribosomal protein 15 were modified by both Ub and SUMO and these modifications displayed divergent abundance alterations upon proteasome inhibition (Fig. 5e, f). These results indicate that reciprocal modification by SUMO and Ub on the same lysine within a target protein may be a relatively rare event, but that it clearly does occur at some sites in the proteome and that these modification events may impart differential functional outputs for the substrate proteins. These results also point to a unique strength of our method in that it allows for parallel mapping of regulated ubiquitylation and SUMOylation sites from a single sample.

**Identification of SUMOylation in tissue samples**. One advan-tage of our method is that it allows for the identification of endogenous SUMOylation sites from native tissues without exo-genous SUMO protein expression. To establish this application of our method, we processed murine tissue from brain, heart, muscle, and liver to establish this application of our method, we processed murine SUMOylated proteins in vivo. We identified 144 unique SUMOylation sites across the four tissues (Supple-mentary Data 2). Overall, muscle and liver had the most similar SUMOylated proteins, sharing ~70% of the total sites observed in those tissues, whereas brain had the most unique SUMOylated proteins that were not found in any other tissue type. We iden-tified the well-characterized SUMOylated protein RANGAP1 in all tissues validating our approach. Interestingly, we again iden-tified ubiquitin as a SUMO-modified protein with lysines 48 and 63 serving as the SUMOylation sites in all tissues. This result suggests that ubiquitin is SUMOylated at critical lysine residues in vivo and validates that our approach can be successfully applied toward the identification of endogenous SUMOylation sites in tissues.

## Discussion

We set out to develop a method for global profiling of native SUMOylation events by taking advantage of the propensity of WaLP for cleavage after threonine. By simply substituting WaLP for trypsin, it was possible to immunopurify and identify a large number of KGG-containing peptides corresponding to SUMO remnants. A large number of the identified sites corresponded to as-yet-unreported SUMOylation events. Several reasons could explain the large number of new sites identified. First, as we reported previously, the orthogonal specificity of WaLP allows cleavage of proteins at sites that may not be accessible to tryp-sin[21]. Second, although previous studies attempted to achieve

minimal expression of their mutant SUMO construct, it is pos-sible that slight overexpression of SUMO or the presence of mutant sequences could cause unnatural SUMO attachment. Third, our method does not differentiate between SUMO1–4, whereas Hendriks et al. examined only SUMOylation sites uti-lizing SUMO-3 attachment[18]. One caveat is that WaLP also cleaves after Leu and to some extent Ile, so the method does not distinguish between SUMOylated, Fat10ylated, or Fub1ylated proteins. However, the observation that 88% of WaLP diGly proteins were reduced upon SENP1/2 treatment argues that the vast majority of observed KGG sites arise from SUMOylation. Additionally, analysis of the observed sites recapitulates previous reports of the expected motifs of SUMOylation and GO term enrichment[19].

A powerful advantage of our method is that it allows for simultaneous determination of ubiquitylation and SUMOylation in the same sample. The same population of cells or tissue can be subjected to analysis of both Ub attachment and SUMO attach-ment simply by splitting the sample in two and digesting half with trypsin and the other half with WaLP. The samples can then be processed in parallel to immunopurify peptides for the presence of the KGG modification, and sequence by mass spectrometry. For the mass spectrometry, it is best to use optimized ionization approaches and data analysis tailored to the non-tryptic WaLP peptides. Although the commercial antibody used in this study was developed for enrichment of KGG-peptides from Ub mod-ification, it appears to also efficiently immunopurify KGG-peptides from SUMO modification. Finally, the method allows for identification of SUMOylated proteins under native conditions including from tissue samples.

## Methods
**WaLP**. WaLp was expressed from Lysobacter enzymogenesis type 495 (ATCC) using Bachovichin's media supplemented with MEM vitamins and 60 g l$^{-1}$sucrose. L. enz. was grown at 30 °C with shaking at 100 rpm for 3 days. WaLP was purified from the culture supernatant as described previously[33]. Briefly, the protease was captured from the supernatant by batch binding on SP-sepharose, which is washed extensively and then eluted with high pH glycine buffer. After buffer exchange to pH 7.2, the enzyme was loaded onto an FPLC monoS 10/10 column and eluted with a gradient of 10 mM NaHPO$_4$, pH 7.2 to the same buffer containing 250 mM sodium acetate over 1 h.

**SENP1 and SENP2 deSUMOylation**. To verify the in vitro deSUMOylation and deubiquitylation assay, untreated HCT116 cells were harvested and lysed with denaturing lysis buffer (8 M urea, 150 mM NaCl, 50 mM Tris pH 7.8, 1 mM sodium orthovanadate, 1 mM NaF, 1 mM sodium 2-glycerophosphate, protease inhibitor tablet (Roche), 5 mM N-Ethylmaleimide (NEM, made fresh in methanol). Lysates were sonicated and centrifuged at 20,000×g for 10 min at 4 °C to remove insoluble material. Lysates were then diluted to 1 M urea using cold 50 mM Tris pH 7.8. Dithiothreitol (DTT) was then added to the lysates at a final concentration of 15 mM. Lysates were then untreated or treated with a mixture of SENP1/2 (Life Sensors) or Usp2cc (Enzo Life Sciences) for 4 h at 25 °C. For SENP1/2 treatment, we initially used 2 U mg$^{-1}$, 4 U mg$^{-1}$, and 6 U mg$^{-1}$ concentrations for the vali-dation experiments and 1 µg mg$^{-1}$ (total protein) for Usp2cc. Reactions were quenched by addition of SDS sample buffer and samples were then processed for SDS-PAGE and immunoblotting. The antibodies used for immunoblotting were against ubiquitin (MAB1510, EMD Millipore, 1:1000 dilution), SUMO1 (4930, Cell Signaling Technologies, 1:1000 dilution), and SUMO2/3 (4971, Cell Signaling Technologies, 1:1000 dilution).

Hela cells were cultured in MEM media with 10% fetal bovine serum and 1% penicillin/streptomycin at 37 °C with 5% CO$_2$. Heat shock was performed at 43 °C for 1 h, then Hela cells were washed with cold PBS, and harvested with 8 M urea lysis buffer (50 mM Tris pH 8.0, 8 M urea, 1 mM vanadate, 2.5 mM sodium pyrophosphate, 1 mM beta-glycerol-phosphate). Extracts were sonicated and centrifuged at 20,000×g for 15 min to remove insoluble material. Protein concentrations were measured by Bradford Assay.

The cell lysate was diluted fourfold by addition of 50 mM Tris buffer pH 8.0 to a final urea concentration of 2 M, and a DTT stock solution was added to a final concentration of 4.5 mM. Equal amounts of lysates were reduced with 4.5 mM DTT at 56 °C for 30 min and alkylated by adding iodoacetamide to 9 mM for 15 min in the dark. Additional DTT was added to lysate to concentration of 4.5 mM. In vitro deSUMOylation reaction was performed by adding specific SUMO protease 1 (SENP1) and SUMO protease 2 (SENP2) (Life Sensors, Malvern,

PA) at 10 units per milligram of lysate protein and incubating with lysate for overnight at room temperature. In vitro de-ubiquitination reaction was performed by adding USP2cc (Enzo Life Sciences) at an enzyme-to-substrate ratio of 1:500 and incubating at RT for 4 h. Untreated control lysate samples were also prepared. Efficiency of SUMOylation cleavage by SUMO proteases and ubiquitin cleavage by USP2cc was verified by western blots using primary antibodies for SUMO1 (4930), SUMO2/3 (4971), and ubiquitin (3933) from Cell Signaling Technology. The lysates were digested by WaLP or trypsin, an enzyme-to-substrate ratio of 1:100 at 37 °C overnight with slow rotation. Digestion was stopped by adding 20% TFA solution to final TFA concentration of 1%. Peptides were then subject to C18 cleaning by Sep-Pak cartridges (Waters Corp.) and lyophilized.

**Enrichment of KGG-containing peptides**. For each 5–7 mg peptide sample, 10 µl of UbiScan beads (20 µl of slurry, Cell Signaling Technology) was used to immunopurify the KGG-containing peptides according to the CST protocols. First, the digests were resuspended in 0.35 ml 2× IAP buffer (100 mM MOPS, 20 mM $Na_2HPO_4$, 100 mM NaCl, pH 7.5), the pH was adjusted to 7.5, and cleared by centrifugation at 10,000×$g$ for 10 min at 4 °C. Then the digests were pre-cleared by rocking with protein A resin for 1 h at 4 °C. Next, the KGG-peptides were immunopurified from the digests by incubation with the UbiScan beads for 2 h with rocking at 4 °C. After isolating the beads by centrifugation at 1000×$g$ for 1 min, the beads were washed two times with 1× IAP buffer, then four times with HPLC-grade water. The peptides were eluted in two steps. The beads were incubated at room temperature with 55 µl of 0.15% TFA for 10 min, centrifuged at 3500×$g$ for 1 min, and the supernatant carefully saved. Then the beads were incubated for an additional 10 min with followed by 45 µl of 0.15% TFA for 10 min, centrifuged at 3500×$g$ for 1 min, and the supernatant was combined with the first elution. The samples were analyzed by nLC-MS/MS on an orbitrap Fusion or LUMOS for WaLP-digested samples, and on a Q-Exactive for Ub-digested samples.

**MS analysis of in vitro deSUMOylation experiment**. Immunoprecipitated peptides were resuspended in 0.125% formic acid and analyzed by an Orbitrap Fusion™ Lumos™ Tribrid™ mass spectrometer (Thermo) coupled to an EASY-nLC 1200 (Thermo). Each sample was split and analytical replicate injections were run to increase the number of identifications and provide metrics for analytical reproducibility of the method. Standard peptide mix (MassPREP™ Protein Digestion Standard Mix 1, Waters) was spiked in each sample vial in a total quantity of 100 fmol (33 fmol per injection) prior to LC-MS/MS analysis. The sample was loaded onto an EASY-Spray™ analytical column (PepMap™, 75 µm × 50 cm, C18, 2 µm, 100 Å, Thermo), which was connected to an EASY-Spray™ ionization source (Thermo). The column was heated to 45 °C for all runs. Mobile phase solvent A was composed of 0.1% formic acid and water. Mobile phase solvent B was composed of 0.1% formic acid, 93.5% acetonitrile, and water. Peptides were separated using a gradient from 5% B to 32% B over 90 min and continued to 53% B over 5 min at a constant flow rate of 300 nl min⁻¹. Full MS scans were obtained with a range of $m/z$ 300–1500 at a mass resolution of 120,000 ($m/z$ 200), with an AGC target value of 4.0E5 and maximum injection time of 50 ms. To select peptides for MSMS analysis, ions with charge states from 2 to 7 were all included, dynamic exclusion was set to 60 s with mass tolerance 10 ppm, and intensity threshold was set at 2.0E3. Data-dependent mode was established at top speed of 3 s. Most intense precursor ions were selected and isolated with a window of 2 $m/z$ and fragmented by collision-induced dissociation with a normalized collision energy of 35 and activation Q of 0.25. MS/MS spectra were acquired in the ion trap at enhanced scan rate with an AGC target value of 3.0E3 and maximum injection time at 350 ms. Real-time recalibration of mass error was performed using lock mass[34] with a singly charged polysiloxane ion $m/z =$ 371.101237.

MS/MS spectra were evaluated using SEQUEST and the Core platform from Harvard University[35–37]. Files were searched against the Swissprot homo sapiens FASTA database updated on September 2015. A mass accuracy of ±5 ppm was used for precursor ions and 0.02 Da for product ions. Enzyme specificity was limited to trypsin, with at least one tryptic (K- or R-containing) terminus required per peptide and up to four mis-cleavages allowed. No enzyme specificity was restricted for WaLP-digested samples. Cysteine carboxamidomethylation was specified as a static modification, oxidation of methionine residues was allowed, and digly remnant on lysine residue (+114.0429) was allowed for each enrichment sample set. Reverse decoy databases were included for all searches to estimate false discovery rates, and filtered using a 1% FDR in the linear discriminant module of core. Results were further narrowed by mass accuracy based on clustering of forward and reverse assignments in Xcorr vs. mass error plots. All quantitative results were generated using Progenesis V4.1 (Waters Cooperation) and Skyline Version 3.1 to extract the integrated peak area of the corresponding peptide assignments.

Quantitative comparison of the KGG-containing peptides before and after SENP1/2 treatment was performed using Progenesis V4.1 (Waters Cooperation) and Skyline Version 3.1 to extract the integrated peak area of the corresponding peptide assignments according to previously published protocols (19, 20). Extracted ion chromatograms for peptide ions that changed in abundance between samples were manually reviewed to ensure accurate quantitation in Skyline. Statistical analysis of the quantitative data was done using two-tailed $t$-test between

SUMO proteases treated and untreated groups. The maximum negative log-$p$ value from three comparison pairs was used to indicate significance for abundance changes of a certain peptide between two groups. Bar graphs of the quantitative data were generated and clustered in Spotfire Decision Site (TIBCO Software AB) version 9.1.2.

**SILAC quantitation of Ub and SUMO**. HCT116 cells were grown on DMEM SILAC media (Thermo Scientific/Pierce) supplemented with 10% FBS, penicillin, streptomycin, arginine (85 mg l⁻¹), and either $^{13}C_6^{15}N_2$ lysine (Cambridge Isotope Labs) or unlabeled lysine at 50 mg l⁻¹. Cells were expanded up to twenty 15 cm plates for heavy medium and twenty 15 cm plates in light medium to yielded 40 mg of total protein. The cells were treated with 10 µM MG132 (Sigma) dissolved in DMSO for 4 h or DMSO-only as the negative control. Cells were then washed with PBS, scraped into ice-cold PBS, and counted with a TC20 cell counter (Bio-Rad). Equal quantities of unlabeled and labeled cells from each condition were combined and stored at −80 until lysis.

Frozen cell pellets were thawed quickly and resuspended in 4 mls of denaturing lysis buffer containing 50 mM Tris, pH 8.2, 8 M urea, 75 mM NaCl, 1 mM $Na_3VO_4$, 1 mM β-glycerophosphate, 1 mM NaF, 2 mM NEM, 1 mM PMSF, and Roche complete mini protease inhibitor. Cells were then sonicated on ice using 15W power output for three cycles of 30 s with 30 s rests in between. Insoluble material was precipitated by centrifugation at 20,000×$g$ for 15 min at 4 °C and protein in the supernatant was quantified by BCA assay. Typically, 5–10 mg protein was digested for Ub identification and 10–15 mg protein was digested for SUMO identification. For Ub identification, lys-C was added to a final concentration of 10 ng µl⁻¹ and incubated for 2 h at 37 °C. Then the digest was diluted to 2 M urea by addition of 50 mM Tris-HCl, pH 8.2 and trypsin (Sigma) was added to a final ratio of 1:100 and incubated at 37 °C overnight. For SUMO identification, the lysate was diluted to 2 M urea by addition of 50 mM Tris-HCl, pH 8.2 and WaLP was added to a final ratio of 1:100 and incubated at 37 °C overnight. Each reaction was stopped by acidification with TFA to 1% (v/v) and clarified by centrifugation at 20,000×$g$ for 10 min at 4 °C. Peptide solutions were then desalted using tC18 Sep-Pak (either 200 or 500 mg, Waters) as previously described[38] and lyophilized.

For the SILAC experiments, initial purification of the peptides was performed using basic pH reverse phase chromatography on a 100 mm × 10 mm ID bridged-ethylene hybrid (BEH) C18 column with 5 µm particles (Waters) in 10 mM ammonium formate pH 10. Using a flow rate of 3 ml min⁻¹ and a gradient of 0–100% 10 mM ammonium formate in 90% ACN over 1 h, thirty-two fractions were collected and every fourth fraction was pooled to obtain four fractions from each digest. After lyophilization and resuspension in 0.5% TFA, the peptides were desalted again with Sep-Pak tC18 cartridges (50 or 200 mg size), lyophilized, and stored at −80 °C.

**Preparation of tissue samples**. Murine tissue samples such as brain, heart, muscle, and liver were obtained from mature BALB/c mice (Cell Signaling Technology). Tissue was homogenized in 8 M urea lysis buffer (50 mM Tris pH 8.0, 8 M urea, 1 mM vanadate, 2.5 mM sodium pyrophosphate, 1 mM beta-glycerol-phosphate). Lysate was reduced by 4.5 mM DTT for 30 min at 55 °C. Reduced lysate was alkylated with 10 mM iodoacetamide for 15 min at 25 °C in the dark. Sample was diluted fourfold with 50 mM Tris, pH 8, and digested overnight with WaLP at weight ratio of 1:100 at 37 °C overnight with slow rotation. Digested peptide lysate was acidified with 20% TFA to a final concentration of 1%, and peptides were desalted over 360-mg Sep-Pak Classic C18 columns (Waters, Milford, MA). Peptides were eluted with 40% acetonitrile in 0.1% TFA and lyophilized.

**nLC-MS/MS for SILAC experiments**. For the SILAC samples, WaLP digest data were collected on an Orbitrap Fusion mass spectrometer (Thermo Scientific) equipped with a Proxeon Easy nLC 1000. Samples were resuspended in 8 µl of 5% formic acid/5% acetonitrile and were loaded onto a 100 µm inner diameter fused-silica micro capillary with a needle tip pulled to an internal diameter less than 5 µm. The column was packed in-house to a length of 35 cm with a C18 reverse phase resin (GP118 resin 1.8 µm, 120 Å, Sepax Technologies). The peptides were separated using a 120 min linear gradient from 3 to 25% buffer B (100% ACN + 0.125% formic acid) equilibrated with buffer A (3% ACN + 0.125% formic acid) at a flow rate of 600 nL min⁻¹. Precursor spectra were collected with a target resolution of 120,000 in the Orbitrap using a scan range of 300–2000 $m/z$. The top 10 precursors with intensity greater than 5000 were fragmented sequentially with CID and ETD in the ion trap with the rapid scan rate, resulting in two separate spectra for each selected precursor ion.

For SILAC samples after WaLP digestion, two searches for each file were performed, one specifying fixed light lysine and one specifying fixed heavy lysine. All searches allowed variable oxidation of methionine, variable protein N-terminal methionine loss and acetylation at alanine or serine, variable peptide N-terminal pyro-glutamate from Q, variable KGG, and fixed modification of cysteine (sample dependent). Following the conversion of the raw data to the open format .mzXML using Proteowizard[39], peptides were identified by database search (2015 Uniprot reviewed human proteome) with MS-GF+ trained for peptides from WaLP digestion as described previously[21]. Current releases of MS-GF+ (2017-01-13) are available at https://omics.pnl.gov/software/ms-gf with development and support at https://github.com/sangtaekim/msgfplus. CID/ETD pairs were analyzed together

when available from the Fusion data. The .mzid output from fixed heavy lysine database searches was processed using R scripts[40] to combine the mass of heavy lysine and diGlycine into one modification. R scripts used in this work are available at https://github.com/komiveslab/SUMO. All .mzid files were then converted to pepXML for compatibility with TPP[41] using idconvert.exe (Proteowizard[39]). PeptideProphet was used to refine heavy and light identifications separately[42]. iProphet was used to combine files corresponding to HPRP fractions from a single condition, and to combine the results from separate heavy and light database searches[43]. PTMprophet was used to generate localization scores for KGG[41,44]. SUMO-remnant-containing peptides were then filtered to 1% FDR by probability score, and ptm sites with localization scores below 0.9 were removed.

The trypsin-digested SILAC samples were analyzed on a Q-Exactive as previously described[14] and data were analyzed with the Core platform from Harvard University[36,37].

To quantify changes in SUMOylation or ubiquitylation upon MG132 treatment, the ratio of the peak areas for the heavy vs. the light were calculated and when protein modifications were identified by multiple peptides, the weighted average of the ratio was computed. The log(2) of each ratio was plotted and sites with log(2) ratios outside 1 s.d. from the median were flagged as "changing." The proteins containing these sites were then analyzed for GO term enrichment with metascape (http://metascape.org/gp/index.html#/main/step1) using all human proteins as background.

**Data availability**. All data, spectra, and result files are accessible via FTP at: Public FTP link: massive.ucsd.edu/MSV000081018—Massive ID: MSV000081018. Proteome exchange ID: PXD006398. The data available on the web interface reflects the MS-GF+ output, before the 1% FDR filter was applied in TPP. The data that support the findings of this study are also available from the corresponding author on request.

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

## Acknowledgements

This work was supported by NSF MCB 1244506 (E.A.K.), NIH T32 EB009380 (J.G.M.), NIH T32 GM 008326 (R.J.L.), New Scholar awards from the Sidney Kimmel Foundation

for Cancer Research and the Ellison Medical Foundation, a Hellman Fellowship, and a NIH New Innovator Award (DP2 GM119132) (E.J.B.).

## Author contributions

E.A.K. conceived the study. R.J.L., A.S.A., and E.J.B. performed all mass spectrometry-based methods and analysis. H.G., Y.Z., and M.L. performed the in vitro SENP1/2 treatment and subsequent KGG enrichment and LC-MS/MS analysis. R.J.L., E.A.K., and E.J.B. performed data analysis. J.G.M. and R.J.L. developed custom scripts to facilitate computational analysis. K.R.C. assisted in spectra validation and computational support. The manuscript was written by E.J.B. and E.A.K. with assistance from R.J.L.

## Additional information

**Competing interests:** H.G. and Y.Z. are employees of Cell Signaling Technology. The remaining authors declare no competing financial interests.

