## [Peer Review File · Nature Communications]

Reviewers' comments:

Reviewer #1 (Remarks to the Author):

Analysis of post-translational modifications using mass spectrometry is a powerful method to obtain global insight into signal transduction. Excellent methodology is available to study phosphorylation, acetylation and ubiquitylation at the endogenous level and site-specific. In contrast, studying ubiquitin-like modifications at the endogenous level and site-specific is challenging. Whereas multiple site-specific methods have been published, these methods use exogenous SUMO, albeit expressed at close to endogenous levels. An endogenous method has been published, but the identification of SUMO sites was not included (Becker et al. 2013 Nat Struct Mol Biol). The authors have developed a novel method to deal with this challenge. They modify the method developed to study ubiquitylation (Kim et al. 2011 Mol Cell) by using the same antibody against di-glycine attached to lysines in target proteins and an alternative protease WaLP. Overall, the approach is interesting, but the poor specificity of WaLP could lead to the false identification of ubiquitylation sites as SUMO sites. This needs to be addressed more thoroughly as detailed below.

1 The WaLP protease has been available for many years, but has never become popular in the mass spectrometry community, because of lack of specificity. In figure 1b, the cutting preference of WaLP is summarized. The enzyme preferentially cuts after alanine, serine, threonine and valine, but also frequently after methionine, glycine, cysteine, leucine and isoleucine. The frequencies for cleavage after other amino acids are lower and for some amino acids, the enzyme is suggested to not cut at all: after proline, glutamic acid and arginine. However, the enzyme still cuts infrequently after aspartic acid and lysine. The cutting frequencies need to be shown in considerably more detail. These numbers don't represent percentages as the numbers don't add up to 100%. What are the real cutting frequencies that are now depicted as 0 or 1? Given the tens of thousands of peptides generated upon proteolytic cleavage of proteomes from higher eukaryotes, even low frequency cleavages after certain amino acids by WaLP can result in a considerable number of corresponding peptides. The cleavage specificity numbers are derived from an earlier publication by the same group as mentioned by the authors (figure 3, Meyer et al 2014 Mol Cell Proteomics 13:823-835). In this earlier paper, it is clear that the enzyme still cuts after arginine at a low frequency. This raises the important issue that a subset of the identified "SUMO" sites in reality could represent ubiquitylation sites, a major disadvantage of the method, especially given the high abundance of ubiquitylation in cells compared to sumoylation. The authors mention that at most 8% of the identified sites involve cleavage after arginine, which is considerably higher than the "0" mentioned in figure 1b. The authors could make their method much more stringent by first purifying SUMO conjugates, using the previously published method (Becker et al. 2013 Nat Struct Mol Biol).

2 To deal with the challenge of false-positive identification of ubiquitin sites as SUMO sites, the authors performed a control by adding recombinant SUMO proteases SENP1 and SENP2 to the lysate, reducing the amount of SUMO conjugates. Subsequently, they find that the majority of identified SUMO sites went down at least two-fold. This control could be carried out more stringently. By increasing the concentration of the SUMO proteases, and by using

these proteases in multiple rounds of treatment, they could virtually completely remove SUMO conjugates. Reduction in sumoylation for the identified targets could then be much higher.

3 To deal with the challenge of false-positive identification of ubiquitin sites, the authors need to include the ubiquitin protease USP2 in the same manner as described for SUMO proteases SENP1 and SENP2. When depleting extracts for ubiquitin conjugates, how many identified "SUMO" sites are reduced?

4 The localization score cut-off used by the authors is > 0.75 . This cut-off is not very stringent and need to be > 0.90 .

5 Concerning subcellular localization of the identified SUMO targets, most of the proteins cluster in the nucleus as expected. However, focal adhesion components and cytoplasmic ribosomal components were also identified, which is rather unexpected, given the dominant localization of SUMO in the nucleus. An explanation for this unexpected finding is missing. Moreover, since ribosomes are mainly assembled in the nucleus, how can one be sure that the reported sumoylation of these ribosomal components occurs in the cytoplasm?

6 A detailed supplemental list of all sites that have been found in the project is missing.

7 In line with the Paris guidelines, it is common practice in mass spectrometry to provide annotated MS/MS spectra for all identified PTM sites as supplemental data. These data are currently missing.

8 The authors combine SENP1 and SENP2 to reduce SUMO conjugates. Since SENP1 has a preference for SUMO1 and SENP2 for SUMO2, they could aim to differentiate between SUMO1 conjugation sites and SUMO2 conjugation sites by using these SUMO proteases separately instead of combined.

9 A good method to map endogenous SUMO sites can be used to study clinical samples or tissue material that can't be studied using the currently available methods. The manuscript could be improved by including at least one sample that can't be studied using exogenous SUMO.

Reviewer #2 (Remarks to the Author):

This manuscript presents a novel approach that allows, for the first time, the large-scale identification of endogenous SUMOylation. This strategy involves the digestion of SUMOylated proteins using the WaLP protease which cleaves proteins at the C terminus of Threonine, Valine, Alanine, Serine and Methionine residues. Upon digestion, SUMOylated proteins will release peptides containing a diglycine-modified lysine at the SUMOylation site that can be enriched using the commercially available K(GG) antibody typically used for ubiquitin remnant immunoaffinity purification.

While this approach is a first step toward the identification of SUMO sites in endogenous

substrates, there are several limitations that need to be addressed to properly evaluate its analytical potentials. The most significant limitation of this approach is the non-specificity of the WaLP protease that gives rise to a yet undetermined number of false positive identifications. The extent of false positive identification is unknown and should be examined in greater detail for this approach to be widely applicable. Additional comments are provided below.

Major comments:

1. NEDD8 and ISG15 have a -RGG at the C terminus, therefore cannot be distinguished from ubiquitylation site upon tryptic digestion, FUBI and FAT10 have respectively a LGG and IGG C terminus that can be cleaved by WaLP (figure 2b peptide is an example). The effects of SENP1 and 2 haven't been tested for these two modifications; therefore the authors cannot exclude the possibility that some sites were wrongly identified as SUMOylated.
2. Line 178: 12% of the GG modified peptides identified were not significantly downregulated after SENP1/2 treatment. Moreover, 85 of those sites (7% of the 1156 GG sites from the WaLP digest) were also identified as ubiquitylated. This raises significant doubts regarding their identification as real SUMO targets. The authors need to provide spot check western blots for some of those proteins to confirm they are not artifacts.
3. Figure 2c: The sequence identified by the author is FGKEFDKHF K^* where K^* is a GG modified lysine and is presumably one of their best case examples of SUMO substrates. However, the two amino acids following the "modified lysine" on the protein sequence are two glycine residues. Due to the lack of specificity of WaLP, this spectra cannot be confidently identified as a SUMOylated peptide and the site should be either confirmed (is the same site identified on another peptide?) or removed.
4. Identified SUMO peptides that lack proper sequence coverage around the modified lysine and that are preceded or followed by two Glycines (2*57.0215) or one Asparagine (1*114.0429) should be validated or removed.
5. The authors must provide a list of all identified SUMOylated peptides sequences with surrounding amino acids (2 amino acids before the N terminus and 2 amino acids after the C terminus) and a list of identified sites for both SUMOylation and ubiquitylation.
6. The authors should provide some statistics about their approach, especially the enrichment efficiency for KGG IP after trypsin and WaLP digestion.

Minor comments:

1. Line 26 and Figure 2a: "of which 931 have not been previously reported". Based on figure 2a, 859 haven't been reported in "Hendricks et al. 2014" and 1084 haven't been reported in PSP. Moreover, Uniprot is currently the database more up to date regarding SUMOylation.
2. Line 91 & 289 & 434: at line 91, the authors say SUMO2, 3 and 4 are targets of WaLP. At line 289, they say their approach cannot distinguish SUMO1, 2, 3 and 4. At line 434, they say their approach is used to map SUMO1, 2 and 3. The authors need to clearly state which paralogs of SUMO are recognized by WaLP.
3. Line 279: "specificity" is not appropriate.
4. Line 289: "isoforms" \diamond "paralogs". Those 4 proteins do not come from the same gene.
5. Line 290: According to Uniprot gene name, Hendriks SUMO mutant is actually SUMO3
6. Line 454: "seven" \diamond "eight"

7. Line 559: The authors should state which experiments were performed using 2DLC. This will provide guidance for people who want to use this strategy in other contexts.
8. Figure 2b and 2c: Only the observed fragments should be drawn on peptide sequence.
9. Figure 2c: one of the lysine is not highlighted in blue.

Detailed response to Reviewers' comments NCOMMS-16-15673-T

Reviewers' comments:

Reviewer #1 (Remarks to the Author):

... Overall, the approach is interesting, but the poor specificity of WaLP could lead to the false identification of ubiquitylation sites as SUMO sites. This needs to be addressed more thoroughly as detailed below.

1. The WaLP protease has been available for many years, but has never become popular in the mass spectrometry community, because of lack of specificity. In figure 1b, the cutting preference of WaLP is summarized. The enzyme preferentially cuts after alanine, serine, threonine and valine, but also frequently after methionine, glycine, cysteine, leucine and isoleucine. The frequencies for cleavage after other amino acids are lower and for some amino acids, the enzyme is suggested to not cut at all: after proline, glutamic acid and arginine. However, the enzyme still cuts infrequently after aspartic acid and lysine. The cutting frequencies need to be shown in considerably more detail. These numbers don't represent percentages as the numbers don't add up to 100%. What are the real cutting frequencies that are now depicted as 0 or 1? Given the tens of thousands of peptides generated upon proteolytic cleavage of proteomes from higher eukaryotes, even low frequency cleavages after certain amino acids by WaLP can result in a considerable number of corresponding peptides. The cleavage specificity numbers are derived from an earlier publication by the same group as mentioned by the authors (figure 3, Meyer et al 2014 Mol Cell Proteomics 13:823-835). In this earlier paper, it is clear that the enzyme still cuts after arginine at a low frequency. This raises the important issue that a subset of the identified "SUMO" sites in reality could represent ubiquitylation sites, a major disadvantage of the method, especially given the high abundance of ubiquitylation in cells compared to sumoylation. The authors mention that at most 8% of the identified sites involve cleavage after arginine, which is considerably higher than the "0" mentioned in figure 1b. The authors could make their method much more stringent by first purifying SUMO conjugates, using the previously published method (Becker et al. 2013 Nat Struct Mol Biol).

The reviewer points out two important issues here

1) WaLP is a non-specific protease. Since we already published the cleavage propensity, we have removed Fig. 1B. We now discuss in the paper that the amount of cleavage after Arg varied because of small amounts of contaminating trypsin from scale-up of the anchorage-dependent cells. We now explicitly point this caveat out in the paper and have performed additional experiments to directly address this issue. We now show data in which we treat lysates with the deubiquitylating enzyme Usp2cc and examine the impact on the SUMO and Ub-modified proteome. While 97% of ub-modified peptides decreased in abundance upon Usp2cc treatment, less than 2% of KGG-peptides generated by WaLP decreased in abundance. This result, along with the previous data with in vitro SENP1/2 treatment strongly suggest that the WaLP digestion and subsequent immunoaffinity purification using KGG-antibodies isolates SUMOylated and not ubiquitylated peptides. 2) We agree with the reviewer that we could have made the method more stringent by using an immunopurification step, but these are also prone to challenges particularly from whole proteome samples, and we wanted to keep the method as simple as possible. We believe the additional SENP and Usp2cc experiments presented in this revised manuscript address the concerns of specificity. No evidence was found for mis-assignment of Ubs for SUMOs in the method.

2 To deal with the challenge of false-positive identification of ubiquitin sites as SUMO sites, the authors performed a control by adding recombinant SUMO proteases SENP1 and SENP2 to the lysate, reducing the amount of SUMO conjugates. Subsequently, they find that the majority of identified SUMO sites went down at least two-fold. This control could be carried out more stringently. By increasing the concentration of the SUMO proteases, and by using these proteases in multiple rounds of treatment, they could virtually completely remove SUMO conjugates. Reduction in sumoylation for the identified targets could then be much higher.

To address if higher amounts of SENP1/2 enzymes would result in more complete deSUMOylation, we treated cell lysates with increasing concentrations of SENP1/2 and determined the impact on global SUMOylation by immunoblotting. All concentrations of SENP1/2 resulted in equivalent removal of SUMO1 and SUMO2/3 conjugates. This data is represented in new figure 3A. Note that even the high concentration used in this study (6U/mg) is less than what was originally used in the initial SENP1/2 proteomic experiment (10U/mg). Based on this result we determined it would not be worthwhile to perform large-scale, and costly, quantitative proteomic experiments using higher concentrations of SENP1/2 that would not substantially alter our findings.

3 To deal with the challenge of false-positive identification of ubiquitin sites, the authors need to include the ubiquitin protease USP2 in the same manner as described for SUMO proteases SENP1 and SENP2. When depleting extracts for ubiquitin conjugates, how many identified "SUMO" sites are reduced?

This was an excellent suggestion and we have now performed this important control and the results were as expected and detailed in our response to point #1. All data from Usp2cc treatment is not presented in new figure 3C. We thank the reviewer for this suggestion!

4 The localization score cut-of used by the authors is > 0.75 . This cut-of is not very stringent and need to be > 0.90 .

We changed the cut-off as recommended.

5 Concerning subcellular localization of the identified SUMO targets, most of the proteins cluster in the nucleus as expected. However, focal adhesion components and cytoplasmic ribosomal components were also identified, which is rather unexpected, given the dominant localization of SUMO in the nucleus. An explanation for this unexpected finding is missing. Moreover, since ribosomes are mainly assembled in the nucleus, how can one be sure that the reported sumoylation of these ribosomal components occurs in the cytoplasm?

Previous studies have also identified ribosomal proteins as SUMOylation targets so we wouldn't necessarily consider this a surprising result. Due to the fact that we generate whole cell lysates prior to immunoaffinity isolation of SUMO-modified peptides, we lose all cellular spatial resolution. The reviewer is correct in noting that we cannot know if SUMOylation of ribosomal proteins is happening in the nucleolus as part of ribosome biogenesis or on actively translating ribosomes in the cytoplasm. This information can be gleaned from performing cellular fractionation steps prior to isolation of SUMOylated peptides. While this would certainly result in interesting findings, this type of experiment is outside the scope of this study.

6 A detailed supplemental list of all sites that have been found in the project is missing.

We apologize for this oversight as this table was submitted to MassIVE but we didn't provide a separate excel table to reviewers. We now provide all identified sites in supplemental table 1.

7 In line with the Paris guidelines, it is common practice in mass spectrometry to provide annotated MS/MS spectra for all identified PTM sites as supplemental data. These data are currently missing.

All data has been uploaded to the MassIVE archive (https://massive.ucsd.edu/ProteoSAFe/result.jsp?task=09831580207e44ce90c54fca37d593e1&view=view_result_list). The files can be accessed via FTP at:

<ftp://MSV000081018@massive.ucsd.edu>, Password: alphalytic) and all annotated MS/MS spectra are available within this data submission.

8 The authors combine SENP1 and SENP2 to reduce SUMO conjugates. Since SENP1 has a preference for SUMO1 and SENP2 for SUMO2, they could aim to differentiate between SUMO1 conjugation sites and SUMO2 conjugation sites by using these SUMO proteases separately instead of combined.

For proof of concept, we wished only to remove as much SUMOylation as possible. We think that differentiating SUMO1 versus SUMO2/3 conjugation sites is certainly an interesting application of our method but this is beyond the scope of the paper.

9 A good method to map endogenous SUMO sites can be used to study clinical samples or tissue material that can't be studied using the currently available methods. The manuscript could be improved by including at least one sample that can't be studied using exogenous SUMO.

We thank the reviewer for this great suggestion. We now demonstrate that our method can identify in vivo SUMOylation sites from mouse tissues. These sites are now included as supplemental table 2.

Reviewer #2 (Remarks to the Author):

This manuscript presents a novel approach that allows, for the first time, the large-scale identification of endogenous SUMOylation. This strategy involves the digestion of SUMOylated proteins using the WaLP protease which cleaves proteins at the C terminus of Threonine, Valine, Alanine, Serine and Methionine residues. Upon digestion, SUMOylated proteins will release peptides containing a diglycine-modified lysine at the SUMOylation site that can be enriched using the commercially available K(GG) antibody typically used for ubiquitin remnant immunoaffinity purification.

While this approach is a first step toward the identification of SUMO sites in endogenous substrates, there are several limitations that need to be addressed to properly evaluate its analytical potentials. The most significant limitation of this approach is the non-specificity of the WaLP protease that gives rise to a yet undetermined number of false positive

identifications. The extent of false positive identification is unknown and should be examined in greater details for this approach to be widely applicable. Additional comments are provided below.

Major comments:

1. NEDD8 and ISG15 have a -RGG at the C terminus, therefore cannot be distinguished from ubiquitylation site upon tryptic digestion, FUBI and FAT10 have respectively a LGG and IGG C terminus that can be cleaved by WaLP (figure 2b peptide is an example). The effects of SENP1 and 2 haven't been tested for these two modifications; therefore the authors cannot exclude the possibility that some sites were wrongly identified as SUMOylated.

The reviewer is correct, we cannot distinguish FUBI and FAT10 modifications using WaLP cleavage. Eventually, researchers may wish to add an anti-SUMO IP step. That said, we showed that ~90% of the modifications we observed were reduced by SENP1/2 treatment so certainly the majority of the sites are, in fact, SUMO modification.

2. Line 178: 12% of the GG modified peptides identified where not significantly downregulated after SENP1/2 treatment. Moreover, 85 of those sites (7% of the 1156 GG sites from the WaLP digest) were also identified as ubiquitylated. This raises significant doubts regarding their identification as real SUMO targets. The authors need to provide spot check western blots for some of those proteins to confirm they are not artifacts.

The reviewer is correct, there was a population of modified peptides that were not reduced by SENP1/2 treatment. Of these peptides, 45% were found previously reported to be ubiquitylated, which is slightly higher than the previously reported percentage of SUMOylated lysines that are also found ubiquitylated (30%). While this does remain a concern we further addressed this by adding an experiment using in vitro Usp2cc treatment to remove all ub-modified peptides from the lysate and evaluate its impact on the SUMO-modified proteome using our WaLP-based method. As described in the manuscript and in response to reviewer #1, 97% of ub-modified peptides decreased in abundance upon Usp2cc treatment, and less than 2% of KGG-peptides generated by WaLP decreased in abundance. These data strongly suggest that we identify SUMOylated and not ubiquitylated peptides using our WaLP approach. We attempted to "spot check" some of our identified SUMOylated proteins using endogenous IP and western blotting, but all antibodies we obtained were insufficient for IP. It should be noted that for this to work, a significant (at least 5%) portion of the protein would need to be SUMOylated for us to be able to observe it by IP-western blot approaches. We anticipate that the majority of our identified SUMOylation targets will be modified at very low stoichiometry.

3. Figure 2c: The sequence identified by the author is FGKEFDKHF K^* where K^* is a GG modified lysine and is presumably one of their best case examples of SUMO substrates. However, the two amino acids following the "modified lysine" on the protein sequence are two glycine residues. Due to the lack of specificity of WaLP, this spectra cannot be confidently identified as a SUMOylated peptide and the site should be either confirmed (is the same site identified on another peptide?) or removed.

We thank the reviewer for pointing this out. We have removed all such peptides from the dataset and we replaced the spectrum in Fig 2c with a more representative sequence.

4. Identified SUMO peptides that lack proper sequence coverage around the modified lysine

and that are preceded or followed by two Glycines (2*57.0215) or one Asparagine (1*114.0429) should be validated or removed.

We thank the reviewer for this suggestion and have now removed all such peptides from the dataset.

5. The authors must provide a list of all identified SUMOylated peptides sequences with surrounding amino acids (2 amino acids before the N terminus and 2 amino acids after the C terminus) and a list of identified sites for both SUMOylation and ubiquitylation.

We apologize for not including these data in the original submission. We have included them as Supplementary Tables 1 and 2 in the revision.

6. The authors should provide some statistics about their approach, especially the enrichment efficiency for KGG IP after trypsin and WaLP digestion.

The enrichment efficiencies varied between 10 and 40%. They depended on the amount of starting material more than on the enzyme. It was more dependent on how fresh the antibody beads were.

Minor comments:

1. Line 26 and Figure 2a: “of which 931 have not been previously reported”. Based on figure 2a, 859 haven’t been reported in “Hendricks et al. 2014” and 1084 haven’t been reported in PSP. Moreover, Uniprot is currently the database more up to date regarding SUMOylation.

We have included comparisons to the 2016 Hendriks paper as well as to Uniprot in the revised manuscript

2. Line 91 & 289 & 434: at line 91, the authors say SUMO2, 3 and 4 are targets of WaLP. At line 289, they say their approach cannot distinguish SUMO1, 2, 3 and 4. At line 434, they say their approach is used to map SUMO1, 2 and 3. The authors need to clearly state which paralogs of SUMO are recognized by WaLP.

We thank the reviewer for catching this mistake, WaLP does not distinguish any of the SUMO isoforms and have corrected all instances noted by the reviewer

3. Line 279: “specificity” is not appropriate.

We changed specificity to propensity.

4. Line 289: “isoforms” “paralogs”. Those 4 proteins do not come from the same gene.

We have made this correction

5. Line 290: According to Uniprot gene name, Hendriks SUMO mutant is actually SUMO3

The reviewer is indeed correct. Despite the Hendriks paper noting that they tagged SUMO2, the sequence that they provide is actually SUMO3. As such we have made this correction in our manuscript.

6. Line 454: “seven” “eight”

We have made this correction.

7. Line 559: The authors should state which experiments were performed using 2DLC. This will provide guidance for people who want to use this strategy in other contexts.

We have now added this information. It was only the SILAC experiments that used 2DLC.

8. Figure 2b and 2c: Only the observed fragments should be drawn on peptide sequence.

We have made this correction.

9. Figure 2c: one of the lysine is not highlighted in blue.

We have made this correction.

**

REVIEWERS' COMMENTS:

Reviewer #2 (Remarks to the Author):

The authors have appropriately addressed concerns raised in my first report, and I recommend publication of the revised manuscript.